# Evaluation of Fracture Resistance and Microleakage of Endocrowns with Different Intracoronal Depths and Restorative Materials Luted with Various Resin Cements

**DOI:** 10.3390/ma12162528

**Published:** 2019-08-08

**Authors:** Ouqba Ghajghouj, Simge Taşar-Faruk

**Affiliations:** Department of Prosthodontics, Faculty of Dentistry, Near East University, Nicosia 99138, Cyprus

**Keywords:** endocrown, CAD/CAM materials, resin cements, fracture resistance, microleakage

## Abstract

The aim of this study was to determine the effect of restoration design on the fracture resistance of different computer-aided design/computer-aided manufacturing (CAD/CAM) ceramics and investigate the marginal leakage of endocrowns according to different types of cement. In total, 96 extracted mandibular first premolars were used for fabrication of endocrowns; 48 of the endocrowns were divided into 6 groups (n = 8) according to intracoronal cavity depth (2 and 3 mm) and CAD/CAM ceramics (lithium disilicate IPS e.max-CAD, zirconia-reinforced glass-ceramic Vita Suprinity, and poly-ether-ether-ketone (PEEK)). Teeth were subjected to a fracture resistance test with a universal test machine following thermo-cycling. Failure modes were determined by stereomicroscope after the load test. The rest of the endocrowns (n = 48) were produced by Vita Suprinity ceramic and divided into 6 groups (n = 8) according to the cement used (Panavia V5, Relyx Ultimate, and GC cement) with intracoronal cavity depths of 2 and 3 mm. Microleakage tests were performed using methylene blue with stereomicroscope after thermo-cycling. Numerical data for both fracture resistance and microleakage tests were obtained and evaluated by three-way ANOVA. PEEK endocrowns had higher fracture resistance compared to lithium disilicate and Vita Suprinity. Panavia V5 cement had the lowest degree of microleakage, while GC cement had the highest. Different intracoronal cavity depths had no correlation with fracture resistance and microleakage.

## 1. Introduction

Caries, physical trauma, abrasions, and erosion can lead to severe loss of tooth tissue and, may also require endodontic treatment. However, excessive loss of tooth tissue and canals that have been treated can often affect the prognosis of the tooth. Endodontically treated teeth may be subjected to intense stress under functional forces; fractures in these teeth are often observed [1]. Post-core restoration is among the preferred restoration protocols of teeth with endodontic treatment. Traditionally, post and core restorations were used to stabilize remaining coronal tooth structures [2]. Throughout recent developments in dentistry, there is still controversy about cases of restoration where the tooth root is weak due to endodontic treatment. In addition, a non-natural structure is formed in the teeth that does not resemble tooth tissue and is restored with a post made from different materials [3]. On the other hand, the main cause of failure is the loss of excess tissue occurring in the tooth structure during endodontic treatment and preparation of the post cavity. Especially in cases where the roots are thin, the amount of tooth tissue remaining after preparation for post-treatment decreases and the risk of fracture may increase [4]. For these reasons, there are alternative treatments, such as ceramic inlay, ceramic onlay, and endocrown restorations in cases of non-vital teeth and substantial tissue loss [5]. 

In 1999, Bindle and Mörmann described endocrowns for the first time as adhesive endodontic total porcelain crowns fixed to endodontically treat posterior teeth [6]. Macro-mechanical retention is provided by the pulpal walls, so endocrown restorations is be fixed to the internal portion of the pulp chamber and on the cavity margins, and micro-mechanical retention is obtained by the use of adhesive resin cement [7]. Furthermore, the fact that endocrowns are one-part restorations is a great advantage over restorations produced by the conventional post-core approach. Endocrowns produced with a minimally invasive preparation can be prepared and simulated more simply than others, because they do not contain multiple technical steps, such as placement of the post into the root canal and shaping of the core structure [8]. 

The long-term success of endocrowns depends on many factors, including proper case selection, correct preparation, and choice of appropriate ceramic and bonding agents [9]. In addition, endocrowns have significant features, such as better aesthetics and mechanical performance, lower cost, and shorter production time than the traditional methods [10]. Restorations of endodontically treated teeth should provide the ideal function, and the aesthetics of these teeth with excessive material loss and the restoration should maintain the remaining tooth structure without any marginal microleakage [11]. 

The depth of the cavity (intracoronal extension), which affects the retention and stabilization of endocrown restorations, can also have an effect on the internal cavity volume, cavity surface area, and marginal-internal adaptation [12]. However, there is little data on the effect of cavity depth on fracture resistance [9]. Furthermore, the effect of cement type on microleakage in endocrowns is still controversial. Although there are studies [5,6,7,8,9,10,11,12] investigating the fracture resistance of endocrown restorations produced by using various computer-aided design/computer-aided manufacturing (CAD/CAM) blocks in the literature [13], there is no comparative study evaluating the effect of cavity depth on fracture resistance for endocrown restorations produced using different CAD/CAM materials. Therefore, the purpose of this in-vitro study was to compare the fracture resistance of endocrown restorations with two preparation designs produced by different CAD/CAM materials and to evaluate the effect of three types of cement on microleakage for endocrown restorations. The null hypothesis of this study was that various CAD/CAM materials and preparation designs would not differ in fracture resistance, and that the type of resin cement and preparation design would have no influence on microleakage.

## 2. Materials and Methods

### 2.1. Specimen Preparation

For the experiment, 96 human mandibular first premolars were collected within 4 months in the Department of Oral and Maxillofacial Surgery at Near East University and stored in distilled water at 37 °C. Prior to use, teeth were washed under running water to eliminate storage solution remnants. The study was conducted in accordance with the Declaration of Helsinki, with approval by the Research Ethics Committee of Near East University (approval no.: 56-540/2018).

Teeth were positioned perpendicularly to the acrylic resin, less than 3 mm below the cement-enamel junction, to remove the tooth crown and pulp chamber. Square acrylic molds with a length of 1.5 cm were used for this procedure. In this study it would be desirable to have a single root canal and have the same measurements (mesio-distal width, bucco-lingual thickness, and crown height).The dimensions of all selected teeth were standardized nearly similar at the cemento-enamel junction; (buccolingually: 7.2 ± 10 mm and mesiodistally: 5.0 ± 0.5 mm) and root length of 13 ± 1.0 mm, determined by a digital caliper. The inclusion criteria for the samples were a completed root formation, absence of root visible fracture lines and carious lesions. Then the radiographic examination was done. Standardization of roots was determined by X-ray (Meditrix, Dentalkart, New Delhi, India). An access cavity was prepared with a diamond-coated stainless-steel bur (Hager and Meisinger GmbH, Berlin, Germany). Pulp tissue was removed with an endodontic reamer (Hager and Meisinger GmbH, Berlin, Germany). In order to ensure standardization of all teeth, the same person carried out the root canal treatment stages in the same order. Removal of pulp chamber was accomplished by using a round carbide bur (Hirschensprungstr.2, Jota AG, Rüthi, Switzerland). The crown-down technique was used to prepare root canals with the help of the Protaper Universal NiTi rotary system (Dentsply Maillefer, Ballaigues, Switzerland), following the manufacturer’s recommendations (Dentsply, Tulsa, New York, USA). As an irrigation solution, 1% sodium hypochlorite was used for 10 s between each file for root canal expansion. Root canals were obturated with thermo-plasticized gutta-percha (DiaDent Group International, Seocho-dong, South Korea). Obturation was performed by an electric motor (X-Smart Dentsply Maillefer, Ballaigues, Switzerland) to ensure standardization. Then flowable resin composite (Filtek Z350XT Flowable, 3M ESPE, St. Paul, MN, USA) was used to fill the canals up to the level of the pulp chamber. After completion of teeth preparation, teeth were stored at 37 °C in distilled water for 15 days.

### 2.2. Teeth Preparation for Endocrowns

The occlusal surfaces of all samples (n = 96) were reduced at least 2 mm in the axial direction; this reduction was achieved by drilling 2 mm deep grooves as guides, then using a diamond wheel bur (Hager and Meisinger GmbH, Berlin, Germany) to reduce the occlusal surface. The bur was oriented along the major axis of the tooth and held parallel to the occlusal plane to ensure a flat surface. A standardized cavity preparation was performed for all teeth, limited to removal of undercut areas of the pulp chamber and alignment of its axial walls with an internal taper of 8–10°. Internal tapering was done by using a tapered diamond-coated stainless-steel bur with a rounded end (Hager and Meisinger GmbH, Berlin, Germany) held perpendicular to the pulpal floor. All internal line angles were rounded and smoothed. The margins of all teeth were further prepared for a 1 mm chamfer finish line, and a 2 mm circumferential ferrule was formed at axial walls. Gutta-percha was removed to a depth not exceeding 2 mm to take advantage of the saddle-like anatomy of the cavity floor. Removal was done with a nonabrasive instrument to maintain the integrity of the root canal entrance.

### 2.3. Fracture Resistance Test

Samples (n = 48) that would undergo the fracture test were prepared according to the cavity depth (intracoronal extension) extending from the central fossa to the cavity base, divided into 2 groups: 2 mm depth (n = 24) and 3 mm depth (n = 24). The preparation depth was standardized with a silicone stopper. A silicone stopper was positioned on the drill to obtain cavity depth. The depth was measured with a digital caliper (Kumpas Digital Prof Metal 150 × 0.01, Digital Caliper, EAGems, Beijing, China) adjusted to within 0.01 mm and with a graded periodontal probe. Each group was further divided into three subgroups according to the CAD/CAM restorative material. Endocrowns were produced with lithium disilicate (IPS e.max CAD, Ivoclar Vivadent Inc., New York, USA) (n = 8); zirconia-reinforced glass ceramic (Vita Suprinity, Bad Säckingen, Germany) (n = 8); and poly-ether-ether-ketone (PEEK) (CopraPeek Light, Whitepeaks, Essen, Germany) (n = 8). All teeth with completed preparations were screened with the help of an intraoral digital scanner (InLab SW15; Denstply Sirona, New York, USA) and the digital data were obtained. The restoration form in the CAD/CAM software (InLab SW15; Denstply Sirona, New York, USA) was selected as a crown for the mandibular first premolar and the biogeneric reference option was selected and transferred to the CAD program (InLab SW15; Denstply Sirona, New York, USA). Afterthe endocrown design was completed (cement range was set to 60 micrometers; minimal occlusal thickness was set to 400 micrometers), data of the 3-dimensional designswere recorded in standard file format (STL) and sent to the CAM unit (InLab MC X5; Dentsply Sirona, New York, USA). Milling of endocrowns using different CAD/CAM blocks was performed. 

Then all endocrowns were cemented with the same dual-cure resin cement (Panavia V5; Kuraray Noritake Dental Inc., Okayama, Japan) under pressure. Pressure was applied by a specially designed apparatus (Figure 1) consisting of an iron base and a rectangular area rising from the middle of the base. The function of the rectangular area was fixation of the sample. There was a hole at the top section of the device with a rod that carried a specific weight. That weight was applied to the sample for 60 s.

Tooth primer (Kuraray Noritake Dental Inc., Okayama, Japan) was applied to prepared tooth surfaces for 20 s to moisten the dentin, then gently air-dried. Ceramic primer (Clearfil, Kuraray Noritake Dental Inc., Okayama, Japan) was applied to the inner surface of the restoration. Cement pastewas mixed and applied to the crown. After 3–5 s of light curing, excess material was removed, then the cement was polymerized by applying light for 40 s to each surface. Samples were stored in distilled water at 37 °C for 24 h. Specimens were then subjected to thermal cycling (5–55 °C, 10,000 times) in a thermal cycling simulation device (MTE 101; MOD Dental, Esetron Smart Robotechnologies, Ankara, Turkey) for 10 s in each bath. The samples were compared in terms of fracture resistance following the thermo-cycle. For this purpose, a universal test machine (EZ-test-500 N, Shimadzu, Kyoto, Japan) was used. The loading piston was centered along the long axis of the specimen with a 6 mm diameter steel ball (Figure 2). The thrust speed of the machine was 0.5 mm/min. Specimens which were attached to stainless steel jig, load was applied until fractures formed. The fracture load was recorded in Newtons (N). In addition, fracture mode was examined for each specimen under stereomicroscope (Leica S8 APO; Leica Microsystems, GmbH, Wetzlar, Germany) and classified according to the following descriptions:Type I: Cohesive failure (in the restorative material of endocrown or in enamel/dentin);Type II: Adhesive failure between the endocrown material and dentin;Type III: Mixed fracture (in both the endocrown material and the dentin).

### 2.4. Microleakage Test

Regarding the second stage of the study, 48 samples were evaluated for microleakage. For this purpose, the groups were prepared according to the cavity depth/intracoronal extension from the central fossa to the cavity base: 2 mm depth (n = 24) and 3 mm depth (n = 24). The depth was measured with a digital caliper (Kumpas Digital Prof Metal 150 × 0.01) adjusted to within 0.01 mm and graded with a periodontal probe. At this stage, only zirconia-reinforced glass ceramic (Vita Suprinity, Bad Säckingen, Germany) was used to produce endocrowns. All prepared teeth were screened with the help of an intraoral digital scanner and the digital data were obtained. The groups, each with eight samples, were as follows: Group I: Dual polymerized adhesive resin cement system (Panavia V5; Kuraray Noritake Dental Inc., Okayama, Japan); Group II:Dual-polymerized adhesive resin cement (Relyx Ultimate Clicker; 3M ESPE, Berlin, Germany); and Group III:Dual-polymerized adhesive resin cement (G-CEM Link Force cement; GC Corporation; Tokyo, Japan). 

Cementation for all groups was held under pressure for 60 s. Self-adhesive resin cements were used as follows: Tooth primer was applied to moist dentin of prepared tooth surface for 20 s, then gently air-dried. Ceramic primer was applied to the inner surface of the restoration. Cement pastewas mixed and applied to the crown. After 3–5 s of light curing, excess material was removed, then light was applied for 40 s to each surface.

Then all samples were stored in distilled water at 37 °C for 24 h to allow maturation of the interfacial bonding [14]. The specimens were subjected to thermal cycles (5–55 °C, 10,000 times) in a thermal cycling simulation device (MTE 101; MOD Dental, Esetron Smart Robotechnologies, Ankara, Turkey) for 10 s in each bath. Afterwards, all samples were immersed vertically downward in a solution of 2% methylene blue dye for 24 h at 37 °C [14]. Then all samples were washed, and buccolingual sections were taken with a slow-speed device (Top Dent, Edenta Golden, Rüthi, Switzerland) by a diamond disc bur (Schrock and Kimmel GmbH, Berlin, Germany) under water cooling. The specimens were rinsed under running water and then dried with tissue paper. Dye penetration was measured (in millimeters) at the tooth–luting agent interface at either the buccal or lingual margins of each surface from the finish line under a stereomicroscope (Leica S8 APO; Leica Microsystems GmbH, Wetzlar, Germany) at 40× magnification. Dye penetration for each tooth was calculated by the average of all readings of the two surfaces.

### 2.5. Statistical Analysis

Statistical analysis of all obtained data from both fracture resistance and microlekeage tests was done by using a statistical software program (SPSS, IBM Statistics 23.0, Chicago, IL, USA). Descriptive statistics including means and standard deviations were calculated for each group. The Shapiro–Wilk test was used to test the normal distribution of the data. Variance analysis was performed with three-way ANOVA, since the data were suitable for normal distribution; t-tests were applied for each group; and Tukey’s post hoc test was performed to compare significant differences between groups. Statistical significance was considered as *p* < 0.05.

## 3. Results

Mean values and standard deviations of fracture resistance for different CAD/CAM blocks of the endocrown restorations are presented in (Table 1). PEEK had the highest fracture resistance value, and e.max had the lowest fracture resistance. Three-way ANOVA revealed that there was a statistically significant difference between the groups (*p* < 0.05). There was a significantly higher (*p* = 0.00) mean fracture resistance value for PEEK (3026N) when compared to Vita Suprinity and e.max (1784 N and 1196N, respectively). In regard to cavity depth, the 2 mm groups had higher fracture resistance than the 3 mm groups but there was no statistical significance (*p* = 0.34). 

The failure mode of e.max and Vita Suprinity with 2 mm and 3 mm cavity depth was generally cohesive failure, and some mixed fractures were noted. PEEK groups (2 mm and 3 mm) demonstrated mostly mixed fractures, but adhesive failure was observed too (Figure 3).

The means of microleakage values for all groups are shown in (Table 2). GC cement presented higher dye penetration than Panavia V5 and Relyx Ultimate cement (Figure 4). There was a significantly higher (*p* = 0.00) mean value of microleakage for Panavia V5 cement (0.154) compared to Relyx and GC cement (0.163 and 0.595, respectively). Concerning cavity depth, the 3 mm groups had more microleakage than the 2 mm groups, but there was no statistically significant difference (*p* = 0.34).

## 4. Discussion

The aim of endocrown restoration is to maintain the tooth, which is not ideal for single-crown or post-core restoration. Endocrowns take advantage of contemporary developments in ceramic CAD/CAM technologies and various types of resin cement [15]. Fracture resistance tests are commonly performed to give information about the hardness and longevity of recent dental CAD/CAM materials [16].

This study shows a significantly higher fracture resistance value for Vita Suprinity (1784 N) compared to the IPS e.max CAD (1196 N).These results are compatible with another study [16], which reported that Vita Suprinity material under scanning electron microscopy (SEM) showed a homogeneous fine crystalline structure, while IPS e.max CAD revealed a structure with fine-grained needle-shaped crystals embedded in a glassy matrix. Therefore, it was concluded that Vita Suprinity had significantly higher fracture toughness, elastic modulus, and hardness than the IPS e.max CAD. In addition, Hamza et al. [17] reported that Vita Suprinity crowns had higher fracture resistance than the lithium disilicate IPS e.max CAD, and this could be attributed to the composition of the material: The addition of zirconia may increase its strength. 

A chewing-simulation test was not performed in this study, but chewing-simulation increased the fracture resistance of lithium disilicate crowns in a study [18] reporting that the fracture resistance of IPS e.max CAD was higher than that of Vita Suprinity after a chewing-simulation test. The inherent mechanical properties of the tested restorative materials play a vital role in fracture resistance, and that was seen in a study [19] that reported higher fracture resistance of IPS e.max CAD than of Vita Suprinity without thermo-mechanical loading; but IPS e.max CAD had the lowest fracture resistance with thermo-mechanical loading. However, the aging processes used in the present study were quite different from those of the previous studies [18,19], which limits the comparability. 

In the current study, endocrown restorations fabricated from PEEK material had the highest fracture resistance (3026 N), and this result was consistent with another study [20] reporting that the fracture resistance of CAD/CAM milled PEEK fixed prosthetics was much higher than that of lithium disilicate glass ceramic and alumina or zirconia. In addition, the fracture resistance of PEEK endocrowns was found to be better than that of feldspathic porcelain or lithium disilicate under a compressive load [21]. The PEEK matrix allows the coalition of carbon and glass fibers for the development of thermoplastic fiber composites, and the increment of carbon fibers safely increased the hardness and fracture resistance [22]. Besides, PEEK has mechanical, physical, and elastic properties similar to human bone, enamel, and dentin, providing bioactivity for PEEK as a crown [23]. In the present research, PEEK material had higher hardness and elasticity, so the findings of Najeeb et al. [23] are in agreement with our results.

In the present study, no statistical significance was found regarding the preparation of cavity depth, thus the null hypothesis cannot be rejected. Our results are in agreement with a study [24] that compared the fracture and fatigue resistance of endocrown restorations with different endo-core lengths and reported that there was no difference in statistical significance between 3 and 4 mm intracoronal cavity depth. 

However, Lise et al. [25] compared the biomechanical behavior of endocrowns with different intracoronal depths and noted that 2.5 mm depth was more susceptible to failure than5 mm depth. In his study, a 45° oblique load was selected to mimic intraoral conditions, but that load created a large moment of force on the premolar. Furthermore, the applied 45° load was found to be more detrimental, because the stress was not distributed along the long axis of the tooth, but it was rather more concentrated at the cervical area. In addition, Dartora et al. [26] determined that fracture resistance was negatively influenced by the depth of intracoronal extension of endocrowns, and reported that the application of periodic loads and temperature changes led to exposure of the hybrid layer, which may have affected the adhesive layer, consequently accelerating the hydrolysis of unprotected collagen, and this led to decreased bond strength between ceramic and dentin tissues over time. This decrease in bond strength may have occurred because of the decreased contact area between intracoronal extensions of endocrowns inside the pulp chamber relative to the remaining teeth. A chewing simulation was not performed, and the force axis used to mimic oral conditions was at a 45° angle instead of the 90° for the fracture test in the present study, which is why our results contradict those of previous studies [25,26].

There are many factors, such as elastic modulus, that play important roles in determining fracture modes of ceramic materials. With regard to failure modes, cohesive failure was observed for Vita Suprinity and IPS e.max CAD groups, but PEEK groups showed fractures in either the endocrown or dentin. These results are similar to another study [25] showing mostly cohesive fracturing, without any mixed fractures in e.max and monoblock zirconia groups. Therefore, PEEK exposed to bending under a load, and with stresses distributed more evenly, results in mixed fractures, while rigid materials produce stress concentrations at critical areas that might cause cohesive failures in the endocrown material [27].

In the current study, the microleakage test was performed after cementation and there was no relationship between microleakage and intracoronal cavity depth. The present study corresponds to the research of Darwish et al. [28], which concluded that cavity depth preparation had no influence on the internal fit of endocrowns; however, axial cavity walls with different divergences were found to affect internal fit. On the other hand, Gaintantzopoulou et al. [29] noted that increased preparation depth negatively affected both the marginal adaptation and internal fit of the final restoration, and the gap values were higher when a deeper cavity was prepared. However, this result contradicts the present study, where different preparation depths did not affect marginal microleakage. Therefore, increased microleakage, scanning process, software design, milling, and shrinkage have an influence on the fitting accuracy of CAD/CAM restoration. 

Shin et al. [11] reported changes in cavity volume, cavity surface area, and marginal and internal discrepancies according to changes in cavity depth; and analyzed the increased volume and surface area of a 4 mm cavity compared with a 2 mm cavity. Results showed that an endocrown with a 4 mm cavity depth had a larger marginal and internal volume. However, in the present study, there was no difference between groups with different cavity depths relative to marginal microleakage. That is why complete seating of the endocrown restorations was done using CAD/CAM technology, which enabled the arrangement of an equal thickness of cement and enhanced the adaptation of margins for endocrown restorations, because incomplete seating may lead to a high marginal gap, and therefore high marginal microleakage [14].

The purpose of this study was to evaluate microleakage according to the type of cement. Panavia V5 cement was found to have the least microleakage; that result is in agreement with a study by Müller et al. [30], which reported that Panavia V5 cement demonstrated considerably lower water absorption than Relyx Ultimate cement. Panavia V5 cement contains hydrophilic aliphatic dimethacrylate but does not have phosphate or hydroxyl groups, or alkaline fillers, which explains why it attained the lowest water absorption.

Trajtenberg et al. [31] showed that Panavia cement (Kuraray America Inc., New York, NY, USA) had less microleakage than Relyx Unicem (3M ESPE, RelyX Unicem 2, Berlin, Germany) and GC (G-CEM Link Force cement; GC Corporation, Tokyo, Japan) cement at both the enamel and dentin margins, which may be due to differences in the pH of the acidic primers between the two cement monomers. Panavia cement needs a primer on the tooth surface to activate its self-etching capabilities, while Relyx Unicemis encapsulated cement, and does not need any type of priming of the tooth surface for activation of its self-adhesive mechanism. The authors [32] reported that GC cement had higher microleakage because it includes 4-methacryloxyethyl trimellitate anhydride, which leads to bonding by a chelating reaction to calcium ions in apatite, and this cement was applied to dry dentin surfaces, since it is water-based and requires drier dentin surfaces for improved adhesion. In addition, the high molecular weight of the functional monomer may be the reason for the failure of the supposed chemical reaction within a clinically reasonable time, leading to relatively weak bonding potential. Tooth pretreatment with acid, primer, and bonding affects the condition of the dentin and the smear layer [33]. It is interesting to note that Uludag et al. [33] showed that Relyx ARC resin cement (3M EPSE, St. Paul, MN, USA) had lower microleakage than Panavia (Kuraray CO Ltd, Osaka, Japan), and reported that the slow polymerization rate of Panavia may allow more water to diffuse from the vital dentin into the hydrophilic interface between the Panavia primer and the dentin, due to its more hydrophilic nature. In the current study, dye penetration was lower for Panavia because the Panavia primer contains molecules (original methacryloyloxydecyl dihydrogen phosphate monomers) which make durable bonds to hydroxyl apatite, metals, and zirconia, leading to limited microleakage [34].

## 5. Conclusions 

Within the limitations of this study, different cavity depths (intracoronal extensions) did not influence fracture resistance or microleakage of endocrown restorations. Comparing PEEK, lithium disilicate, and Vita Suprinity, PEEK had the highest fracture resistance and properties similar to enamel and dentin, while lithium disilicate had the lowest fracture resistance. Lithium disilicate ceramic has aesthetic properties, but the use of PEEK ceramic material can provide acceptable fracture resistance, due to the inherent property of integrated crack prevention. Comparing cements, Panavia V5 resin cement showed the least amount of microleakage, followed by Relyx Ultimate and GC resin cements. The Panavia V5 adhesive resin cement system was found to be the best cement to restrict marginal microleakage. However, further investigation into the longevity and success of endocrown restorations is necessary.

## Figures and Tables

**Figure 1 materials-12-02528-f001:**
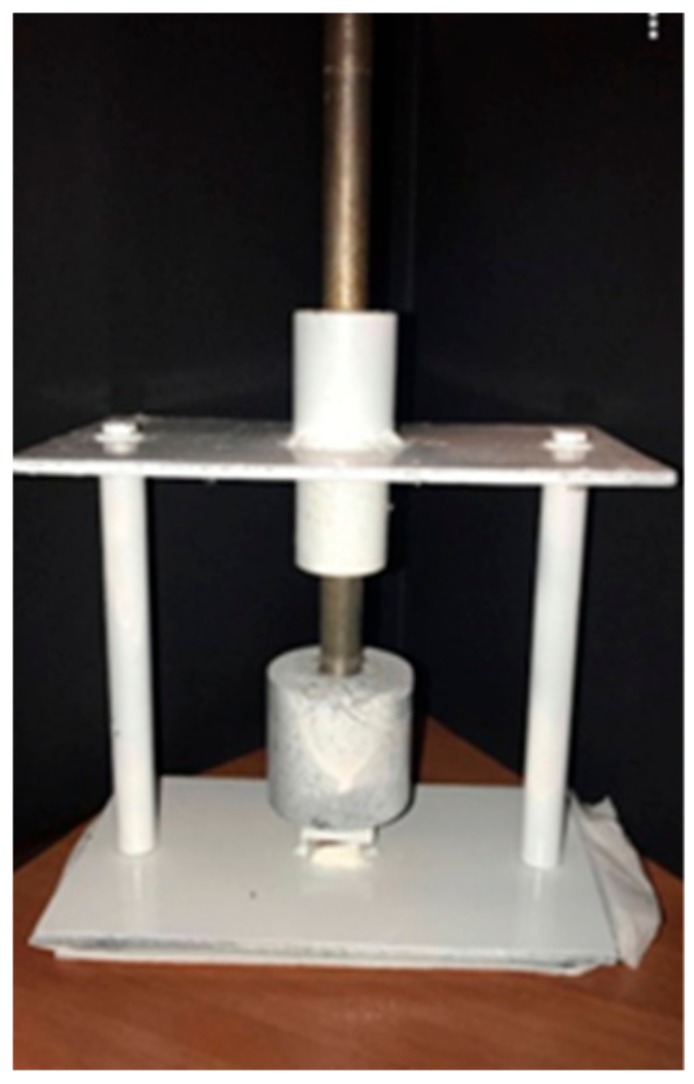
Specially designed device for applying pressure during cementation.

**Figure 2 materials-12-02528-f002:**
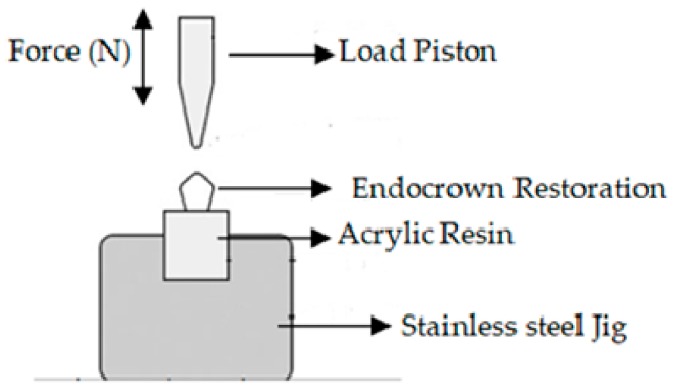
Device for applying fracture resistance test.

**Figure 3 materials-12-02528-f003:**
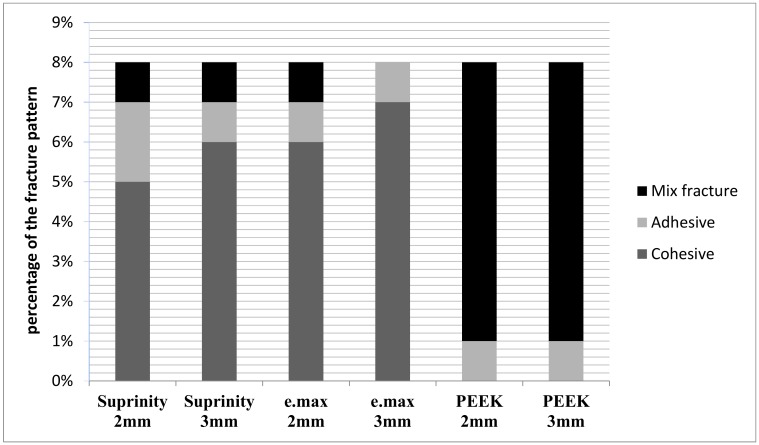
Fracture modes of specimens after the load test.

**Figure 4 materials-12-02528-f004:**
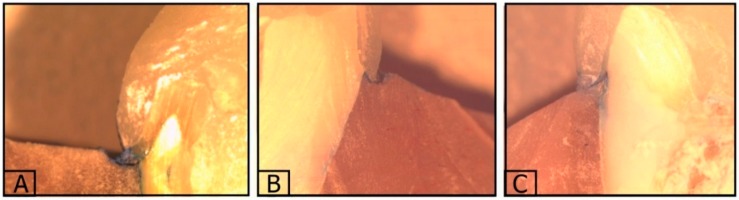
Microleakege evaulation of endocrowns by stereomicroscope: (**A**) Dye penetration for GC cement; (**B**) dye penetration for Panavia V5 cement; (**C**) dye penetration for Relyx Ultimate cement.

**Table 1 materials-12-02528-t001:** Mean values and standard deviations of fracture resistance in Newtons (N) for test groups.

Ceramic Type	Mean	Standard Deviation
Suprinity	1784 N	±226
E.max	1196 N	±303
PEEK	3026N	±270

**Table 2 materials-12-02528-t002:** Values and standard deviations of microleakage for test groups (mm).

Cement Type	Preparation Depth	Mean	Standard Deviation
GC	2 mm	0.595	0.229
	3 mm	0.376	0.131
Relyx	2 mm	0.163	0.151
	3 mm	0.265	0.250
Panavia	2 mm	0.154	0.211
	3 mm	0.145	0.168

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
