# Peer review of "Evaluation of Fracture Resistance and Microleakage of Endocrowns with Different Intracoronal Depths and Restorative Materials Luted with Various Resin Cements"

_materials, 2019, doi:10.3390/ma12162528_

Round 1

Reviewer 1 Report

Unfortunately too many mistakes.

line 13 - 14: cement types on 13 microleakage of endocrown

line 15 endocrown, should be in plural

line 19 rest - THE REST

line 19 - 20:were produced 19 by Vita Suprinity ceramic were divided - repetition!

line 24 compared with - compared to

line 33 - 34 (the) fractures ....

line 38 do not - does not

line 45 - 46: remake the sentence

line 52 more simply - simpler

line 57 - 60: remake the sentence

line 62 - 64: CLARIFY!

line 64: investigate - investigating

line 72: would be no difference....!?

line 82: positioned perpendicular (ly)

line 83 - 84 - remake the sentence

line 86: all these measurements can be done without X-ray!

line 122: in which...!?

line 125: intra oral digital scanner - point out the manufacturer (SIRONA) !?

line 135:...under force pressure for 60 seconds... What type of apparatus did you use. It is not clear enough. Give some more details.

line 155 - 158: repetition of the test explanation!

line 193: ...while the lowest for E-max!?

line 199 - 201: remake the sentence

line 208 - 209: singular or plural!?

277 - 278: remake the sentence

The whole explanation of the results in fig.2 are totally confused.

Reviewer 2 Report

The experiment protocol has to be clarified and improved.

Reviewer 3 Report

1. The paper need English language editing. Sentence structure and phrasing are awkward in many places, for example (lines 58-60) "The restorations of that endodontically treated tooth should be able to provide the ideal restoration, function, and aesthetics for these teeth which have suffered an excessive material loss and it should maintain the remaining tooth structure and not to show marginal microleakage"

2. Many sections of text need to be broken up into paragraphs.

3. In the results and in Table 1 - eliminate the decimal points

4, Figure 2 needs SD error bars.

5. The authors need to comment on the following points 

* how was the tooth size (root length and cross-sectional surface area) standardized ?

* because NaOCl weakens dentin, how was the exposure time for NaOCl used in the canal controlled?

Other than these points, the paper is sound and is a useful contribution.

Round 2

Reviewer 1 Report

Well done. You have followed all proposed recommendations.